# Preschool Children's Reasoning about Sound from an Inferential-Representational Approach

**Leticia Gallegos-Cázares** *, **Fernando Flores-Camacho and Elena Calderón-Canales**

Institute for Applied Sciences and Technology, National Autonomous University of Mexico,
Mexico City 04510, Mexico; fernando.flores@icat.unam.mx (F.F.-C.); elena.calderon@icat.unam.mx (E.C.-C.)
* Correspondence: leticia.gallegos@icat.unam.mx

**Abstract:** The purpose of this article is to present an analysis to identify the reasoning processes and representations that preschool students develop about sound based on the inferential-representational approach. Participants were 18 preschool students between the ages of four and five attending three rural schools located in the Sierra Norte of Puebla, Mexico. Data were obtained through a 14 question semi-structured interview. From children's answers to the formulated questions, an inferential analysis method was applied to identify intentionality, representation elements, sign-material expressions, representations, inferences, and coordination rules in students' constructions. The results show that children build a basic set of epistemic tools to give meaning to their interpretations and can use them as surrogate reasoning to make inferences. This research constitutes the first approximation toward the understanding of preschool children's reasoning forms with an inferential-representational approach and constitutes a new approach that puts forward new referents to analyze students of different ages. We consider that the described results and analysis have implications on science education at this educational level.

**Keywords:** preschool; representations; science education; epistemology

## 1. Introduction

The benefits of including science in preschool classrooms are undeniable. The analysis of how students build their representations is still a fundamental question because it allows us to comprehend how we build physical understanding and give meaning to our surroundings. This knowledge lets us build effective strategies that improve teaching and learning processes in the classroom. The ideas that students develop have been the subject of analysis for several decades. Researchers have covered the fields of physics, chemistry, and biology at all educational levels; however, research in children under eight years old has been infrequent, especially in the field of physics [1].

An example of this is the study of sound ideas. Most works over this topic analyze ideas from high school and college students [2,3] and only a small set of works are focused on children under eight years [4–10]. Table 1 presents a synthesis of some reported works.

The studies described in Table 1 show that students' ideas about sound do not allow them to understand diverse situations in which sound is present in their daily lives. On the other hand, although it is important to understand children's ideas and conceptions, previous research does not contemplate how such ideas are developed, the required steps and reasoning processes.

Research questions: How do children build their representations about sound? What is their reasoning process? Is there a set of a basic nucleus in which they base their inferences? To answer these questions, the objective of this study is to identify the reasoning processes and representations that students develop about sound at a preschool level using as a tool the inferential-representational approach analysis to answer these questions.

**Table 1.** Synthesis of research papers of students' ideas about sound.

| Authors | Ages | Ideas about Sound |
|---|---|---|
| Piaget (1973) | 4–5<br>5–6<br>7–10<br>11–15 | Sound is something that remains in its place.<br>Sound is something that comes out of an object toward the ear and returns to the object.<br>Sound moves in a straight line in all directions.<br>Sound expands and travels through the air. |
| Driver, Squires, Ruschworth and Wood-Robinson (1994) | 4–16 | Sound is produced by the actions that subjects make over objects.<br>Sound is a part of objects.<br>Sound travels through the air. |
| Mazens and Lautrey (2003) | 6–10 | Sound cannot pass through solid matter.<br>Sound passes through materials (if they have holes).<br>Sound passes through materials that are not too strong.<br>Sound is associated with the idea of vibration. |
| Eshach and Schwartz (2006) | 13–14 | Sound has substantial characteristics.<br>Sound can push another object.<br>Sound can be contained inside an object.<br>Sound can be added to another sound. |
| Sözen and Bolat (2011) | Basic education | Sound travels through vibration and causes particles to move.<br>Sound is heard through vibration and matter. |
| Delete for review (2018) | 5–6 | Sound has object properties.<br>Sound is related to vibrations, but it is not possible to explain further. |
| Delete for review (2019) | 4–6 | Sound is within objects.<br>Sound depends on a subject's actions and can be perceived because we have ears.<br>Sound bounces in materials.<br>Sound requires space to propagate. |

## 2. Representation and Reasoning: Basic Assumptions

The present paper focuses on analyzing representations that preschool children build about sound from an inferential-representational approach. This method allows us to analyze representations as epistemic entities that, through their constitutive elements, take subjects to the elaboration of inferences through a surrogate reasoning process that will lead to the construction of the best possible explanation (pleasing for the subject).

Before describing the inferential-representational approach and its implications on the analysis of children's representations, it is convenient to establish some preceding aspects necessary to determine the representations' function and children's reasoning process.

*Origin and Function of Representations*

The analysis of representations has been managed from the epistemological and cognitive approaches. In the epistemological approximations, the representations are related to the possibilities of the subject's interaction with the environment. For Wartofsky [11], the representations are the perception medium that is determined by the action with objects, transforming it into a "virtual action" [11] (p. 194). This basic idea that contemplates that action transforms interaction (action-perception) is present in many of the philosophy representatives [12–14]. The representation is a basic interaction element (possible action, interpretation, and intention) with the phenomenal surrounding.

The representation's epistemological conception coincides with the perspectives of the cognitive analysis. The proposals from this perspective start from initial construction, determined by the body's capabilities over the world [15]. This process has been referred to as "embodiment" [16,17]. These representations are not an image derived from a performed action. They are an element of interpretation and interaction in which aspects such as perception, intention, and goals are involved [16,18]. These aspects make representations to characterize them as a structure or an internal mind system.

Burr and Jones [19] present this position in the following way: "How does a system such the brain manage to use its sensory input to represent the states of affairs in the world [emphasis added]". The brain achieves this by utilizing active sensorimotor predictions, which have high reliability, to represent the world in an action-oriented manner" [19] (p. 597).

The original and primary function of representations is to take action within a subject's environment to conserve their existence; however, as representations move to a more abstract plane (graphic and symbolic object), the finality of the representation goes beyond the conservation of existence and transforms into an intervention with diverse intentionality. Additional to this action-intervention function, the representation has an extensional function in which it can be translated to other contexts or situations beyond the ones that originated the representation, amplifying the possibilities of intervention. Therefore, the intervention in the world also implies a change of it: "But beyond this, perceptual activity is now also shaped to, and helps to shape a new and different world, namely that world which is a cognitive construction, and is embodied in our representation, as theories and models in science, and as pictures in art" [11] (p. 195).

## 3. Embodiment's Role in Preschool Children's Reasoning

Although the construction of representations, which implies a mind–body relation, is a process that takes place in multiple situations [15], it can be considered that in a subject's development a basic or nuclear set is built from the representations that constitute an axiomatic base to interpret and develop actions and inferences over a wide variety of situations from a phenomenon or problem presented by others. In other words, they are a set of resources that act as restrictive and corporal action elements from an interpretation of a real or imagined situation of the environment [20].

There are many conceptualizations about embodiment, and in this paper, we consider the conception of embodiment-cognition as: "Cognition is best explained in terms of embodied representational states that utilize the same mental resources as are involved in perception and the guidance of action" [19] (p. 589). Based on the last section, we can say that representations generated through the process of embodiment (embodiment-representations) constitute basic structures of the cognitive system that allow the subject, willingly, to interpret and act upon his or her surroundings. Under this idea, it is possible to think that preschool children's constructions about an explanation of certain phenomena or problems about daily physical aspects, such as sound, are constructed from a set of basic representations built over their development. This aspect is one of the assumptions of the present research.

## 4. Children's Reasoning

One of the changes in our way of conceiving children's reasoning has to do with the idea that cognition is the result of active manipulation, and therefore, action influences over perception, indicating that perception and action are inseparable [10,21,22]. When children are active participants, it is possible to observe more complex reasoning and understanding that are not necessarily expressed verbally, but rather through other resources such as gestures or the combination of gestures and tools [23,24].

One of the opinions we share with Sherin Krakowski y Lee [25] is that students build their reasoning, and therefore, their explanations when they intend to answer questionings; this has also been expressed in the Possible Partial Models [26] where students construct their reasoning based on a set of constricting and corresponding ideas that come from previous cognitive resources.

These reasoning processes can be more or less sophisticated depending on the function of the elements presented as objects, other representations, and the contextual conditions of the questions [27–31]. On this topic, previous conceptions and knowledge domains also have to be considered, given that they have an essential role in the possibilities of scientific reasoning [32,33]. A good example of how preschool children can integrate

previous conceptions in their reasoning process to solve new situations is presented by Hast [34]. The study of inferences developed around curved and inclined movement in which previous conceptualizations about free fall and horizontal movement are integrated.

There is still much to analyze from the processes of reasoning, as expressed by Sherin Krakowski y Lee: "We need a better understanding of student reasoning processes in the circumstances we study; we must understand and model students reasoning as it occurs in the interview, we employ to study commonsense science" [25] (p. 169). Through this paper, we echo the necessity of developing complementary elements to previous works to find new aspects for the comprehension of the reasoning and representation processes. We focus on the perspective of inferential construction of representations, taking as a starting point the recent development on the science's epistemology. This approach represents a new proposal of analysis of the construction of representations and reasoning, providing new elements to understand the comprehension of physical processes in students and give answers to research-oriented questions.

## 5. The Inferential-Representational Approach

This paper adopts the inferential approach of representations [35] because, in comparison to the structural approach, it presents the following advantages: (a) it does not establish any type of similarity requirement between the represented and the representation; therefore, it does not require symmetries between the represented phenomena and the form of representation making no assumption of the catchment of reality or part of it; (b) it solves the problem of not having the necessity to demonstrate the symmetry, reflexivity and transitivity processes that Suárez [35,36] has demonstrated that the structuralist vision cannot solve; (c) it enables the understanding of representations as fictitious entities; (d) the construction or interpretation of representations depends on the elements that the subject has to carry out a surrogate reasoning from representation denotation elements.

The inferential approach can be characterized by factors such as intentionality (referring to the subject's awareness of the purpose or finality of their representation). Interpretation implies recognizing elements that give meaning to the representation, and surrogate reasoning is based on the representation elements to create a hypothesis or explanation that can be applied to what is intended to be represented [35,37,38].

It is important to emphasize that, in this approach, the established representations (by the subject or others) constitute a tool or epistemic artefact, referring to an intentionally built entity materialized through some medium (object, image, or expressed symbol) and used in multiple ways [30–41]. The epistemic tool implies that a subject can build new representations whose level of functionality will be determined merely by the form of expression and by the possibilities given by the theoretical, empiric, and contextual elements that the subject provides.

## 6. The Inferential Approach as an Analytic Tool

With the use of the inferential approach in analytic terms to establish the children's reasoning processes, it is necessary to define the elements in such representations. From a general framework, we propose that representations must comply with the following:

Intentionality (what is intended to be represented). Every representation intends to represent some part of a directed process toward an end or main goal. As pointed out by some authors [42], even six-month-old children are capable of conceptualizing the intentional state agents that have an orientation toward a final goal.

Forms of expression (sign-material). This implies that every representation has a form and material to be expressed. Images, language, charts, objects, among others, determine different reasoning constrictions of the representation provided elements, allowing the combination or ensemble of different forms of representation with different purposes. The representation of an object's movement in some direction is an example of expression forms.

Interpretation, referring to the implication or meaning of the representation. Every subject establishes an interpretation that attributes a function and meaning to the elements

of their interpretation. The previous example has the elements of a vehicle and an arrow, and each element represents an entity; for example, the drawing represents a specific vehicle, and the arrow represents an action (movement) and direction (orientation). The representation of movement only acquires meaning when the element of an arrow is applied to the vehicle representation (by itself, the arrow could have other meanings, such as strength).

Inferentiality, referring to the surrogate reasoning that allows inferences. The representation elements are used to reason with them, resulting in an inference when applied to real or imaginary objects. In the previous example, a subject can infer the movement direction (left or right), the vehicle speed, or if an element will catch the other.

Rules of coordination. They show how the inferred elements of the representation connect with the represented. The verification of inferences requires the construction of procedures to estimate if they are correct or closer to be correct, such as measuring and observing the phenomena. For example, they measure the vehicle speed.

In synthesis, the representation characteristics are based on the subject's ability to establish every function and characteristic of the representation. Such elements will enhance the subject to build knowledge and explanations in a specific context. Without such context, the representation would not have a way to be used as an epistemic tool [43].

## 7. Materials and Methods

### 7.1. Method of Analysis

Based on the inferential conception framework and the representational construction process, we analyze children's reasoning, explanations, and sound representation. Participants' age is between four and five years old and is between the second and third grade of preschool.

### 7.2. Representations as an Epistemic Tool

The representation is developed as an epistemic tool by children. The construction of the representation requires observing and recognizing the elements of the phenomenon through its processes of everyday experience and embodiment. With the elaborated representation, it is possible to establish inferences through a surrogated process of reasoning. In a correlative manner, the construction of representations will be constituted by its elements derived from previous conceptions that establish their cognitive resources. These resources are mainly determined by the basic conceptions derived from the embodiment and experiences that are recovered during the elaboration of inferences and representations.

Based on children's explanations and inferences, it is necessary to determine the reasoning process and the construction representations. Table 2 synthesizes the proposed method of analysis to determine the representations and inferences from students and shows some examples of students' answers.

Every subject will build their epistemic tool based on the described elements, performing an inferential process. Therefore, every interview from this procedure will be analyzed.

From previous research on children about the topic of sound, see Table 1, it can be foreseen that the representational diversity will be bounded to common elements established by the process of embodiment in preschool children. Based on this assumption, possible sets of representations and similar inferences will be established to determine everyday patterns.

### 7.3. Sample

The present research worked with a no probabilistic intentional sample of 18 children attending preschool between four (four boys, five girls) and five (two boys, seven girls). Children were preschool students from three rural schools located in the Sierra Norte of Puebla, Mexico. The schools had a low socioeconomic level. In Mexico, preschool is the first basic education level and is formed by three school grades. The students from our sample

belong to the second and third grade. They take classes together in the same classroom with the same teacher and participate in the same school activities.

**Table 2.** Factors and the way to determine them based on children's expressions.

| Factor to Determine from Representations | Determination Process |
|---|---|
| I. Intentionality | Questions and answers aligning.<br>Example: "How you can produce sound?"<br>If I hit with a stick |
| II. Elements of interpretation:<br>(a). Embodiment-base ideas<br>(b). Daily experience | Direct expressions of a process perception and involved organs in the perception.<br>Example: "It can be hear because you have ears."<br>Expression that refers to some type of observation, self-observation or from others.<br>Example: "The strong sound can be heard even with the ears covered." |
| III. Sign-materials expressions | Constricted expressions from drawings and objects.<br>Example: "The sound stays in the hose." |
| IV. Representation | Representation elements' integration.<br>Example: "The sound travel in the air." |
| V. Inferences | Expressions that indicate a supposition or explanation.<br>Example: "If something is covered with a box then the sound is trapped." |
| VI. Rules of coordination | Expressions of corroboration.<br>Example: "The sound bounces off the hose because I don't hear it anymore." |

### 7.4. Instrument and Materials

A 14 question semi-structured interview was used. The interview considers three basic topics of sound: the first one refers to how sound is produced; the second topic refers to how sound is perceived; the third topic refers to how sound propagates. The interview was semi-structured, and the questions were conducted in three sections: (a) questions about sound production about their experience (objects from the surrounding and materials, specifically designed for the interview, such as drawings of animals, insects, objects, among others); (b) questions about hypothetical situations (cover the head with a box); (c) questions about experimental situations (marimba formed by three keys, each one from a different material, e.g., metal, rubber and wood, a hose telephone, and a musical triangle that was tied to a couple of earmuffs). Figure 1 shows some of the objects and photos used during the interview.

### 7.5. Procedure

Every interview was applied individually with an approximate duration of 30 min. During the interview, a member of the research group participated, and a cameraman was present. The children's teacher was present as an observer. Every interview had the consent of the children's parents, teachers, and the schools' principal. All interviews were transcribed and analyzed based on the presented framework.

Three researchers analyzed the children's responses in the interviews, two of them specialists in science education and physics, the third being a specialist in science education and cognitive pedagogy. The researchers had a 90% agreement and the difference was resolved by consensus.

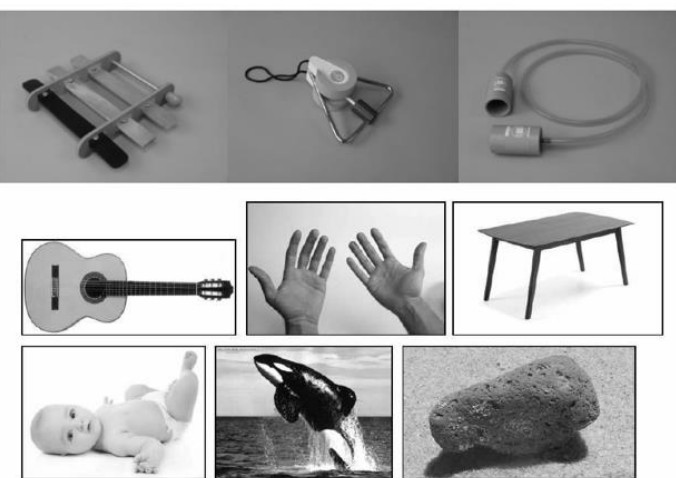

**Figure 1.** Images and instruments used during interviews.

*7.6. Categorisation*

The interviews were transcribed in their totality. An inferential analysis method was applied to the interview transcription from children's answers to the formulated questions. The elements of characterization proposed in the methodology of analysis were identified by the researchers' consensus.

## 8. Results and Analysis

We present the results of every representation factor and reasoning process determined by applying the inferential method.

Intentionality. Eleven children give answers that are aligned to the questions, and they recognize what is being asked about sound and its possibilities of being produced or heard, establishing their representation construction process. Some examples of the answers are: "If I hit with a stick, I can produce a sound", "If you blow it you can produce a sound (trumpet)", "You can listen to it because you have ears", "I speak aloud so he listens because he is on the other side of the school". Seven children seem to confuse the intentionality of the question at the beginning of the interview; therefore, their reasoning is oriented toward communication possibilities. For this case, some examples of students' answers are: "animals do not listen, only if they are from the same species" or "He does not listen because I call him and does not comes". Only one student does not establish any recognition of intentionality, giving diverse answers or is limited to saying, "I don't know".

Elements of interpretation. Throughout the interview, from their elements of embodiment and daily experiences, children assemble the elements that will bring structure to their representation. The directed derived conceptions from the embodiment processes are conceptions derived from the recognition of perception through the ear and the direct action that comes from children's daily experience. These conceptions have been used since the early stages of development. From this condition, children recognized the intentionality of the question and developed expressions that account for their experience through perception–action processes [15,19]. Some of the central or basic elements that preschool children present are the following:

Embodiment-based idea 1 (Emb1). Sound is perceived through the ear, i.e., "It can be heard because you have ears" (100% of the sample).

(Emb2). Sound is produced by an action, i.e., "The sound is produced when you hit the table", "When I speak" (100% of the sample).

(Emb3). More intense sounds (strong) require a major or stronger action, i.e., "If you cover the ears, the sound is strong if you can hear it", "The strong sound can be heard even with the ears covered", "If someone is from afar you have to speak louder so her or she can listen to you" (100% of the sample).

(Emb4). Sound is a substantial or material entity. In diverse ways, children express that sound has material properties; for example, expressions such as "If something is covered, like a box, the sound gets trapped and cannot be heard outside" (two five-year-old girls and one five-year-old boy, one four-year-old girl), "Sound weights and can bring down a tree" (one five-year-old boy).

(Emb5). To continually produce a sound, it is required to maintain an action, i.e., "If you stop hitting a key (from the marimba) or a box, the sound stops", "The sound must be repeated several times so it can be heard, if not it will be lost" (one five-year-old girl).

(Emb6). If there is a space for the sound to pass through, it will be heard. Some representative expressions are: "Sound comes out from everywhere in the box, that's why it can be heard" (one five-year-old child), "Sound passes through the little hole in the hose" (two four-year-old girls and one four-year-old boy, and one five-year-old boy).

Along with these expressions, children establish others that are derived from the observation of others' actions. For example, to recognize that musical instruments produce a sound requires the pulling of chords of a guitar ("stretch") or blowing the trumpet to make a sound. Some other experiences have a mediated interpretation and do not have an immediate recognition as an experience; for example, the notion that different materials produced different sounds. It can be found in an expression such as: "The iron sounds loud", "Metals have strong sound" (one five-year-old girl and one five-year-old boy, one four-year-old boy), "Wood does not produce any sound" (one four-year-old boy and one four-year-old girl).

Sign-materials expressions. These expressions appear in the questions and materials used toward this goal, such as instrument illustrations, animals, objects, and experimental activities such as a hose phone, a triangle with a thread, and earmuffs (see Figure 1). The interpretation that children make with these elements will complete their epistemic tool representation, allowing them to develop their inferences.

Rules of coordination. As previously expressed, these rules have a function to relate the inferences with the real phenomenon; therefore, they are oriented toward proving, which in the case of explanations and predictions in science are usually constituted by measurements and experiments. In the case of young children, we can expect a direct proving such as the affirmation of their explanations by referring to the perceived or observed event. In this case, children express their affirmations by making short expressions such as: "It can be heard", "The sound does not come out", "Yes, the sound is strong", among others.

As it can be appreciated, there is a set of elements where children have to establish a representation of the sound that allows them to infer explanations or possible situations. In the case of specific questions that pose a condition where sign-material elements are present, children build the best possible explanation. For example, one of the activities children did in school was to talk through their hose telephone. A fragment of the following interview illustrates the previously mentioned idea:

I = Interviewer; S1 = student (five years old)

I. Can you listen to the person on the other side of the hose telephone?

S1. Yes, I can.

I. How do you think that sound can reach there?

S1. The noise goes from this tiny hole that you have here until it reaches here. [pointing to the interior of the hose and the sound trajectory]

I. What happened to the sound of my voice? How does it got where you are?

S1. Because it goes like this. [pointing the hose from the middle up to where she is]

I. And what would happen if I fold and squeeze the hose? What will happen when I speak to you again?

A1. I would not listen.

I. You would not listen, why?

S1. Because you are covering it like this, so the noise gets stuck in there, and it will remain there. I cannot hear it.

The previous example shows: how the sound is interpreted as a material or substantial entity (Emb4); how the sound is produced by an action (Emb2); and the notion that sound travels in space (Emb6). These elements assemble with the conditions or sign-materials' expressions of the activity. In this case, the activity consists of two people listening on the opposite sides of a hose telephone. Based on this, the surrogate reasoning and inferences can be constructed as follows:

Sound is a material entity that requires space (hose hole) to travel through and be listened to; if the space is interrupted (folding or squeezing the hose), the sound will not be able to pass through and will be trapped.

To give certainty to the inference, the coordination rule that subject S1 uses is her direct observation on the folded hose and the corroboration that she is not listening through the hose anymore.

From the reasoning process previously shown, it can be noted that the representation used by subject S1 is used as an epistemic tool to elaborate her reasoning that can be summarized as: "The sound is a substantial entity that requires a space to travel". With this reasoning reconstruction process, applied to all participants, the sound representations shown in Table 3 were found.

**Table 3.** Preschool children's sound representations that function as an epistemic tool (T#).

| Epistemic Tools | |
| --- | --- |
| T1 | Sound is a substantial entity that can be perceived through the ear. |
| T2 | Sound is in objects (people) and can be produced by action from them. |
| T3 | Sound is a substantial entity that requires a space to travel. |
| T4 | Sound is produced and transmitted differently depending on the materials. |
| T5 | For a sound to last long, the action that produces it must be repeated. |

These small representations, contextual conditions, and established sign-materials are established for children to solve their situation through reasoning. This allows us to explain how epistemic tools are used and if they can explain children's answers.

In the next section, two examples of different reasoning complexities will be presented.

Example 1. During the interview, a series of drawings (shown in Figure 1) are presented to a four-year-old boy (subject A2).

I = interviewer, S2 = student (four years old)

I. I am going to show you some drawings or photos and you're going to tell me if we can make sound with them (referring to the content of the drawing or photo)

S2. Nod his head.

I. What is this? [showing the card]

S2. A guitar.

I. What could we do with it to make a sound?

S2. Playing it.

I. How would you play it? If the guitar were real, which part would you touch to make a sound?

S2. Stretching the threads.

. . .

I. What is this? [showing a card with the drawing of a spider]

S2. A spider.

I. Do you think a spider can hear?

S2. No, she only catches bees or some little worm that flies toward his web.

In this case, animal drawings (whale and spider) do not show any indication of an auditive system in those animals, prompting children to establish direct inferences such as:

Ears are required to listen (T1) because the animal or object (chicken, whale, stone) does not have ears (sign-material condition) in the graphic representations, and thus those entities cannot hear. The lack of ear identification implies that children express other actions that the organisms can perform, as seen in the spider's case.

A more elaborate case of reasoning is presented in example 2.

Example 2. During the interview, a musical triangle is presented (see Figure 1). The instrument is hanging from two strings that are tied to earmuffs. The triangle is played with a ramrod and heard by the student. Before the action, the student is asked:

I = Interviewer; S3 = Student (five-year-old girl)

I. If you put the earmuffs in your ears and I play the triangle, do you think you could listen to the sound?

S3. No.

I. No, why?

S3. Because the ears are covered while you play and after you take them off and you play again you will listen.

I. If I take off the earmuffs and play it again, I will listen, but if you have the earmuffs on with the ears covered you do not listen.

I. Do we try it?

S3. [student nods with her head]

I. Lean a little bit, so the triangle is not close to your body, okay now I am going to play. What happened?

S3. I listen to the sound. Because you play it, and the sound went like this. [The student points to the triangle and then points to the string that holds it]

I. Oh, because I play it, you think the sound went like this, how did it move?

S3. Through the strings

I. Through the strings. Where did the sound travel?

S3. Up to the ears, it went to my ears

I. It went to the earmuffs and then to your ears. Why do you listen to it when it reaches your ears?

S3. They must have holes. [referring to the earmuffs]

In this case, the devices such as the earmuffs or the strings that hold the triangle act as constrictions of the child's representation. The perceived sign-material elements couple with the epistemic tools. In this case, the student's reasoning can be built into two parts:

r1. Sound is a substantial entity and is perceived by the ear (T1); if I cover my ears, it cannot be heard.

This inference is not fulfilled because the student's coordination element does not occur (not listening to the sound) because she listens to the sound while wearing the earmuffs; this leads to the introduction of new elements—some of them are determined by the material but others are superimposed to the material to satisfy the premise.

r2. The sound is produced and transmitted through different materials (T4), the sound travels through the string up to the earmuffs. As the sound is a substantial entity that can be perceived through the ears (T1) and the sound is a substantial entity that requires a space to travel or move (T3), it must travel through a space that can only happen if the material has space (little holes).

As it can be seen, the epistemic tools that guide the reasoning processes when the expected rules of coordination are not met reinterpret the sign-material conditions, forcing the epistemic tool's function to obtain the best possible explanation, which in this case is translated to a porosity attribute (not observed) in the material. This reasoning that belongs to a five-year-old girl shows us the processes of assembling sound representations as epistemic tools to build complex reasonings to interpret and explain complex phenomena, given that the notions of medium vibrations and their perception as sounds are not yet built by children from those ages and even older children, as shown in previous research about the topic (see Table 1).

## 9. Conclusions

The present study considered the elements of intentionality, representation elements, sign-material expressions, representations, inferences, and coordination rules as elements that characterize preschool children's representation of sound. From the results we can highlight the following aspects:

Children build a basic set of epistemic tools to give meaning to their interpretations and can use them as surrogate reasoning to make inferences.

The student's reasoning complexity depends on the sign-material elements that are present during the inquiries. Some questions implicate a direct or straightforward inference where a representation with an affirmative (perceived ears within the drawing) or adverse condition (perceived absence of ears within the drawing) leads to a direct inference. On the other hand, when there are situations that do not correspond to the student's inferences and the rule of coordination is not confirmed, more representations are assembled (epistemic tools) with the sign-materials aspects. This process took place in the experimental activity cases, where more elements constrict the inferential possibilities. Two of the most relevant examples occur in the reasoning cases around the hose phone and the triangle with the earmuffs.

The basic set of representations that work as epistemic tools (Table 3) also appear in previous research over other populations and environments in children within the same age range or older (up to 10–12 years), indicating that this basic set of tools endures through children's development and continues the generation of the same type of ideas, hypothesis, and conceptions, at least in the case of sound [4–6,8–10].

Based on the inferential conception of representations, the analysis proposal constitutes a new approach that comes from the philosophy of science and puts forward new referents to analyze the construction of representations and reasoning processes from individuals of different ages. This approach is applied to the established scientific knowledge in the disciplines and any interaction with the phenomena, involving the interpretation and explanations people create, including preschool-grade children.

As this approach does not demand a similarity between the representation and what is represented (what is real), it can be applied to any representations that children make in the embodiment processes. Although the representations correspond with daily constructions and common sense, they do not correspond with scientific representations.

When children construct their explanations of how sound is heard, how it generates and how it propagates, they employ the acquired knowledge of previous experiences; for example, what they had heard or what they are looking at the moment, such as the use of images during interviews or specific situations as in using the hose phone. These children's explanations show that their representation involves the brain, the body, the context, and the interactions with these elements. The exposed examples demonstrate that preschool children's expressed constructions are not packages of information and definitions, but dynamic representations that come from a cognitive–sensorimotor interaction.

We consider that the described results and analysis have implications on science education at this educational level. On one hand, it provides clarity over children's reasoning and how they construct the knowledge of their physical surroundings, including sound. On the other, it brings elements for the design of activities used in the classroom's

educational process, based on a basic set of representations. The generated representations at this stage have an important implication over posterior learnings. Many of these have a substantial change resistance given that they enhance adequate reasonings to explain children's surroundings. The described results have utility toward the improvement of the design of external representations, for example, the drawings and images of schoolbooks. These external representations provide epistemic tools for children to reason and interpret their classroom activities.

This research constitutes the first approximation toward the understanding of preschool children's forms of reasoning with an inferential-representational approach. There is much more to inquire and new research lines to be opened to understand how abstract concepts are built, which, at first glance, do not seem to have a perceptual correlation. Future steps could be centered on exploring the possibilities of inferential analysis to explain students' reasoning in other knowledge domains and different cultural settings.

**Author Contributions:** Conceptualization, L.G.-C., and F.F.-C.; methodology, L.G.-C., F.F.-C. and E.C.-C.; writing-review and editing, L.G.-C., F.F.-C. and E.C.-C. All authors have read and agreed to the published version of the manuscript.

**Funding:** This research received no external funding.

**Informed Consent Statement:** Informed consent was obtained from all subjects involved in the study.

**Conflicts of Interest:** The authors declare no conflict of interest.

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
