# Peer review of "Preschool Children’s Reasoning about Sound from an Inferential-Representational Approach"

_education, doi:10.3390/educsci11040180_

Round 1
Reviewer 1 Report
The manuscript is original and contributes to a new methodological perspective on analyzing children’s reasoning about scientific topics, based on sound theoretical and epistemological premises. Some suggested improvements to the article involve:
- An editing of English language to make the text more easily readable.
- A more explanatory and well-coordinated presentation of key concepts used in the theoretical and methodological framework and their interrelations. For instance:
- How are different concepts (e.g., epistemic tools, reasoning, etc.) interrelated? A visual representation (i.e., figure) of their interconnectedness could be helpful.
- Instead of summarizing the theoretical framework previously presented, section 7.2 could be more informative on how these theoretical concepts have been instrumentalized, i.e., transformed and integrated into an analysis framework.
- Table 2 could be more explanatory, perhaps using some examples. Also, there seems to be a missing phrase after II (a).
- Similarly, the title of Table 3 seems to require some amendment: what does "too (T)l" stand for?
- Some more detail could be given on the process followed to ensure inter-rater reliability.
- What does the sentence “For the interview, counting of the described materials was made” (lines 256-7) mean?
- Given the small size of the sample it could be preferable to use frequencies instead of percentages in the presentation of results.
- Literature citations are required in the paragraph corresponding to lines 465-9.
Author Response
The authors thank the reviewers for their valuable comments, questions, and recommendations. There are very valuable to improve the document. We hope that the corrections made fully respond to the reviewer's expectations.
The following describes the adjustments made to the document based on the comments of the reviewers:
The manuscript is original and contributes to a new methodological perspective on analyzing children’s reasoning about scientific topics, based on sound theoretical and epistemological premises. Some suggested improvements to the article involve:
- An editing of English language to make the text more easily readable.
Response: An editing of the English language was carried out.
- A more explanatory and well-coordinated presentation of key concepts used in the theoretical and methodological framework and their interrelations. For instance:
- How are different concepts (e.g., epistemic tools, reasoning, etc.) interrelated? A visual representation (i.e., figure) of their interconnectedness could be helpful.
- Instead of summarizing the theoretical framework previously presented, section 7.2 could be more informative on how these theoretical concepts have been instrumentalized, i.e., transformed and integrated into an analysis framework.
Response: The paragraph that summarized the theoretical framework was replaced by a new one indicating the analysis framework's instrumentalisations. The new paragraph is:
The construction of the representation requires observing and recognising the elements of the phenomenon through its processes of everyday experience and embodiment. With the elaborated representation, it is possible to establish inferences through a surrogated process of reasoning. [205–210]
Also, in table 2, some examples were added.
- Table 2 could be more explanatory, perhaps using some examples. Also, there seems to be a missing phrase after II (a).
Response: The error was amended.
- Similarly, the title of Table 3 seems to require some amendment: what does "too (T)l" stand for?
Response: The error was amended, (T)l, was replaced by (T#)
- Some more detail could be given on the process followed to ensure inter-rater reliability.
Response: The reviewer is correct. To amend, a new paragraph was introduced. The paragraph is:
Three researchers analysed the children's responses in the interviews, two of them specialists in science education and physics, the third being a specialist in science education and cognitive pedagogy. The researchers had a 90% agreement, the difference was resolved by consensus. [256-259].
- What does the sentence “For the interview, counting of the described materials was made” (lines 256-7) mean?
Response: The sentences were eliminated, because indeed is not clear.
- Given the small size of the sample, it could be preferable to use frequencies instead of percentages in the presentation of results.
Response: Frequencies changed percentages.
- Literature citations are required in the paragraph corresponding to lines 465-9.
Response: The references [4–6] and [8–10] were included.
Reviewer 2 Report
Is the content succinctly described and contextualized with respect to previous and present theoretical background and empirical research (if applicable) on the topic?
Yes, the author in presenting the research concept of inferential-representational approach strictly follow the rules of its elaboration. One by one they delineate the research problem, list the components of the problem at hand and consistently analyse each of them in the children's statements.
Are the research design, questions, hypotheses and methods clearly stated?
The article clearly states the aims and research methods. The author do not formulate hypotheses and the questions posed seem not very precise (are they the questions presented in paragraphs 42-45)?
The research method is clear, but the research tool used is briefly described. The presented description does not allow us to determine the order of the questions asked and the moment when the children can experiment or look at the drawings. A more detailed presentation of the used research procedure will allow other researchers to repeat the research and confirm the results. This is especially important in research that deals with a new research area - such as the one described in the article.
Are the arguments and discussion of findings coherent, balanced and compelling?
The clear argument does not raise any doubts. The author carefully formulates the results, referring to previous findings.
For empirical research, are the results clearly presented?
The author presents examples of children's behaviour in which statements of interest to the researcher are revealed, indicative of their reasoning.
Other comments
Page 7, Point: 220. Something is missing in point (a) of Table 2.
Page 12, points: 501-506 (suggestions for further research). The author presents this research as preliminary so it will also be important to confirm the findings gathered on another group of children. Cultural differences are also worth noting. It is possible that children in a setting functioning in a sound-rich environment on a daily basis (e.g. children of parents professionally involved in music) will manifest different examples and explanations of sound. This is important insofar as the author draw attention to children's personal experiences in the article.
Author Response
The authors thank the reviewers for their valuable comments, questions, and recommendations. There are very valuable to improve the document. We hope that the corrections made fully respond to the reviewer's expectations.
The following describes the adjustments made to the document based on the comments of the reviewers:
Is the content succinctly described and contextualized with respect to previous and present theoretical background and empirical research (if applicable) on the topic?
Yes, the author in presenting the research concept of inferential-representational approach strictly follow the rules of its elaboration. One by one they delineate the research problem, list the components of the problem at hand and consistently analyse each of them in the children's statements.
Are the research design, questions, hypotheses and methods clearly stated?
The article clearly states the aims and research methods. The author do not formulate hypotheses and the questions posed seem not very precise (are they the questions presented in paragraphs 42-45)?
Response. To clarify the research questions, some changes were made in the paragraph 41–47. The new one is:
Research questions: How do children build their representations about sound? Which is their reasoning process? Is there a set of a basic nucleus in which they base their inferences? To answer these questions, the objective of this is to identify the reasoning processes and representations that students develop about sound at a preschool level using as a tool the inferential - representational approach analysis to answer these questions. [42-46 lines]
The research method is clear, but the research tool used is briefly described. The presented description does not allow us to determine the order of the questions asked and the moment when the children can experiment or look at the drawings. A more detailed presentation of the used research procedure will allow other researchers to repeat the research and confirm the results. This is especially important in research that deals with a new research area - such as the one described in the article.
Response: Indeed, a more explicit description was needed. To do this, section 7.2 was modified with a new paragraph:
The construction of the representation requires observing and recognizing the elements of the phenomenon through its processes of everyday experience and embodiment. With the elaborated representation it is possible to establish inferences through a surrogated process of reasoning.
Also, some examples were added in table 2, and in section 7.4, a new description was made:
A 14 questions semi-structured interview was used. The interview considers three basic topics of sound: the first one refers to how sound is produced; the second topic refers to how sound is perceived; the third topic refers to how sound propagates. The interview was semi-structured, and the questions were conducted in three sections: a) questions about sound production about their experience (objects from the surrounding and materials, specifically designed for the interview, such as drawings of animals, insects, objects, among others); b) questions about hypothetical situations (cover the head with a box); c) questions about experimental situations (marimba formed by three keys, each one from a different material metal, rubber and wood, a hose telephone, and a musical triangle that was tied to a couple of earmuffs). Figure 1 shows some of the objects and photos used during the interview. [237-248]
Are the arguments and discussion of findings coherent, balanced and compelling?
The clear argument does not raise any doubts. The author carefully formulates the results, referring to previous findings.
For empirical research, are the results clearly presented?
The author presents examples of children's behaviour in which statements of interest to the researcher are revealed, indicative of their reasoning.
Other comments
Page 7, Point: 220. Something is missing in point (a) of Table 2.
Response: The error was amended.
Page 12, points: 501-506 (suggestions for further research). The author presents this research as preliminary so it will also be important to confirm the findings gathered on another group of children. Cultural differences are also worth noting. It is possible that children in a setting functioning in a sound-rich environment on a daily basis (e.g. children of parents professionally involved in music) will manifest different examples and explanations of sound. This is important insofar as the author draw attention to children's personal experiences in the article.
Response: The reviewer is correct. Other contexts and personal situations are relevant to children reasoning and are necessary to investigate. There is no much space to make a deep comment on this aspect. We also decide to add to the end of conclusions:
Future steps could be centered on exploring the possibilities of inferential analysis to explain student’s reasoning in other knowledge domains and different cultural settings.[502-503]